# Regulation of Cerebral Blood Flow Velocity by Transcutaneous Electrical Nerve Stimulation: A Preliminary Study

**DOI:** 10.3390/healthcare12191908

**Published:** 2024-09-24

**Authors:** Eun-Seon Yang, Ju-Yeon Jung, Chang-Ki Kang

**Affiliations:** 1Department of Health Sciences and Technology, Gachon Advanced Institute for Health Sciences & Technology (GAIHST), Gachon University, Incheon 21936, Republic of Korea; eunseon57@gachon.ac.kr; 2Institute for Human Health and Science Convergence, Gachon University, Incheon 21936, Republic of Korea; 3Department of Radiological Science, College of Medical Science, Gachon University, Incheon 21936, Republic of Korea

**Keywords:** transcutaneous electrical nerve stimulation, common carotid artery, pulse wave velocity, cerebral blood flow velocity, neuroprotection

## Abstract

Objectives: An excessive and abrupt increase in cerebral blood flow may cause blood vessel damage, leading to stroke. Therefore, appropriate methods to immediately regulate blood flow velocity are important. Through an analysis of 31 healthy adults, we therefore investigated whether stimulating the common carotid artery (CCA) using transcutaneous electrical nerve stimulation (TENS) could modulate blood flow velocity in the CCA. Methods: Three stimulation intensities (below-threshold, threshold, and above-threshold) were applied in a random order. Blood velocity changes were examined by the measurement of peak systolic velocity (PSV) with Doppler ultrasound before, during, and after TENS stimulation. To evaluate arterial stiffness, pulse wave velocity (PWV) was calculated using CCA diameter, and blood pressure was measured before and after stimulation. Results: PSV changes in the below-threshold level were significant (*p* = 0.028). The PSV after below-threshold stimulation was significantly decreased by 2.23% compared to that before stimulation (*p* = 0.031). PWV showed no significant differences; however, a nonsignificant increase was observed immediately after stimulation only in the above-threshold condition. Above-threshold stimulation can increase vascular tone by activating the sympathetic nerve, possibly triggering vasoconstriction. Conclusions: A decrease in blood flow velocity may not be expected upon the above-threshold stimulation. In contrast, the below-threshold stimulation immediately reduces blood flow velocity, without significantly affecting hemodynamic function, such as arterial flexibility. Therefore, this short-term and low electrical stimulation technique can help to lower vascular resistance and prevent vascular damage from rapid blood flow velocity.

## 1. Introduction

The regulatory mechanisms underlying the maintenance of cerebral blood flow (CBF) are essential for normal brain function. In general, increased CBF can enhance brain function by increasing the supply of glucose and oxygen, whereas decreased CBF is associated with brain diseases, including ischemic stroke and neurodegenerative disorders such as Alzheimer’s disease [1]. Therefore, most studies investigating CBF have aimed to increase blood flow, among which various studies have investigated the efficacy of cerebrovascular stimulation in increasing CBF [2,3]. However, the risks of excessive blood flow velocity have recently been reported. Moreover, in situations where an increase in blood flow velocity is induced, neuroprotection becomes important. Maintaining vascular health and smooth blood flow may be considered as key elements of neuroprotection [4], as blood vessel damage causes CBF to decrease, which can cause brain disease. Reduced CBF is a major cause of cognitive decline, and the prevention of vascular damage is important for maintaining brain health and preventing stroke [5,6].

Ischemic stroke, the most common form of stroke worldwide, occurs when blood flow to a specific part of the brain is blocked, triggering damage to brain cells [7]. When high blood flow velocity damages the blood vessels, blood vessel stiffness can increase, while blood flow velocity is reduced, causing ischemic stroke. Under such conditions, a neuroprotective function is required to limit damage to the brain tissue following ischemic injury, and to prevent the death of viable neurons [8]. One previous related study found that common carotid artery (CCA) stiffness is related to cognitive decline and dementia [9]. This study reported that CCA stiffness affects CBF regulation and cerebrovascular health, regardless of age or increased blood pressure (BP), and that this stiffness could eventually lead to cognitive decline. However, the neuroprotective drugs currently used for therapeutic purposes in stroke patients remain limited because of the difficulty in selecting appropriate patients for clinical trials, as well as the limited treatment time range [8,10]. These results emphasize the need for management or treatment to control blood flow velocity and maintain blood vessel health.

Blood flow velocity plays an important role in blood circulation and vascular health, as it is closely related to biophysical metabolism and vascular thrombosis [11]. Recently, several problems related to excessive increases in blood flow velocity have been reported, indicating that a rapid blood flow velocity increases the load on the heart and blood vessels, possibly leading to hypertension due to cardiovascular diseases [12]. Moreover, hypertension can cause an excessive increase in blood flow velocity, which can damage blood vessels [13]. Vascular endothelial cell damage can further induce turbulent flow (due to vasculitis), obstruct flow, or lead to thrombosis, which may result in stroke [14,15,16,17]. In addition, hypertension has long been considered a risk factor for stroke, with prior research indicating that individuals with BP > 120/80 mmHg are more likely to experience stroke [18], whereas a decrease in BP can effectively reduce the risk of stroke [19]. Maintaining appropriate blood flow velocity is important for vascular health. In addition to several biomarkers closely related to blood flow velocity, including BP and blood viscosity, the flexibility of blood vessels can reduce vascular resistance, which can simultaneously decrease blood flow velocity and increase blood flow volume, making these factors important diagnostic biomarkers of circulatory health [20,21]. New approaches that reduce blood flow velocity without increasing vascular stiffness are required. Electrical nerve stimulation may potentially be a useful approach to achieve this.

Transcutaneous electrical nerve stimulation (TENS) is generally applied to treat neuropathic pain caused by neural diseases in various clinical conditions [22]. Further, it has been well-established that its electroanalgesia effect is related to the autonomic nervous system (ANS), and can also influence the cardiovascular system [23,24,25]. According to previous studies, TENS is effective at treating sympathovagal imbalance, particularly hypertension, which is a major risk factor for cardiovascular disease [25,26,27,28,29,30]. While most of these treatments have been reported to affect the peripheral vasculature, their effect on CBF remains poorly understood. The CCA is a cerebral vascular source that supplies blood to the brain, bifurcating into the external carotid artery (ECA) and internal carotid artery (ICA) [31]. Therefore, control of blood flow velocity in the CCA could modulate the delivery of blood to the brain, thereby effectively controlling CBF.

Additionally, blood vessels are innervated with sympathetic nerves; therefore, an immediate effect involving activation of the sympathetic nerves upon electrical stimulation could be expected [32]. However, the neck area, where the CCA is located, is known to be an uncomfortable region for TENS application, leading to a lack of research. This discomfort stems from the fact that TENS applications typically involve intensities above the sensory threshold, and are sometimes unsuitable for application to the front of the neck. Caution is needed, as stimulating the carotid sinus can lead to an excessive hypotensive response, while laryngospasm could be triggered by laryngeal nerve stimulation [33]. Before using CCA electrical stimulation to effectively regulate CBF, validating safe intervention methods and assessing their effects is necessary. Proper regulation of blood flow velocity can improve brain function [34]. In addition, most CBF studies have been conducted on various cerebrovascular stimulations, with the goal of increasing blood flow; therefore, the risk of an excessively increased blood flow velocity has not been seriously considered. Therefore, in the present study, we propose an effective stimulation protocol that could be safely applied to the neck area to improve CBF by adjusting the intensity and duration of TENS. This protocol can be used to achieve neuroprotective effects. To accomplish this, we used diagnostic ultrasound to clarify the location of stimulation, minimized the side effects of TENS using electrical stimulation over a short period, and compared the effects of each stimulation intensity.

## 2. Materials and Methods

### 2.1. Participants

The sample size of this study was calculated in advance using G*Power 3.1.9.4 software following a pilot study. Information on the peak systolic velocity (PSV) change in response to stimulation in 12 participants was used to calculate the sample size. RMANOVA analysis was performed to compare the PSV before, during, and after stimulation, with the results showing that the average PSV values before, during, and after threshold stimulation were 100.13, 97.657, and 97.265 cm/s, respectively. The partial eta squared of this data was calculated as 0.0585, and the effect size was 0.249. Finally, the sample size was calculated as 28 by applying 0.8 power and α error probability 0.05. Considering a dropout rate of 10%, a total of 31 healthy participants in their 20s were enrolled in this study (see Table 1).

All participants provided written informed consent to participate prior to initiating the experiment. The study was approved by the Institutional Review Board (IRB no. 1044396-202207-HR-146-01) of Gachon University and the World Health Organization International Clinical Trials Registry Platform (WHOICTRP registration number: KCT0009172). All participants were restricted from consuming caffeine or supplements that could affect their blood vessels or ANS for 24 h prior to the experiment, to limit any potential effects. Participants with cardiovascular disease, neurological or musculoskeletal disorders, or pain were excluded from the study. Additionally, participants with skin wounds or sensitivity to stimulation, as well as those in whom attaching the TENS electrode pads was difficult were excluded. Before the experiment, biosignals were measured to confirm whether participants fell within normal ranges (systolic blood pressure [SBP] and diastolic blood pressure [DBP] of less than 120 and 80 mmHg, respectively; and heart rate [HR] of 60–90 beats per minute [bpm]).

### 2.2. Experimental Protocol

Before the experiment, the participants laid on a flat examination bed in the supine position for 5 min to obtain their vital signs and CCA diameters in a resting state. Electrode pads (PROTENS, Bio Protech, Gangwon-do, Republic of Korea) were subsequently attached to the CCA area. We used diagnostic ultrasound to clearly confirm the location of the carotid sinus and body to ensure the safe application of TENS and to eliminate adverse effects related to carotid sinus stimulation, after which electrode pads were then attached at the furthest possible location. In other words, the pads were attached to the CCA near the clavicle, which forms the distal part of the carotid sinus. The electrode pads were wrapped with electrical insulating tape to prevent penetration of the ultrasound gel. Additionally, each pad was cut to a size of 2.7 × 2 cm^2^ for focal stimulation of target blood vessels in order to stimulate the minimum range (see Figure 1). When attaching the electrode pads, care was taken to ensure they did not touch the clavicle, and all were positioned with the cathode (−) at the bottom and the anode (+) at the top. This placement aligns the direction of blood flow with the direction of electron movement, thereby inducing hyperpolarization at the anode location according to anodal blocking theory [30].

The frequency and width for the TENS intervention (ITO, Saitama, Japan) were fixed at 2 Hz and 50 μs with a constant mode, respectively, and the following three levels of stimulation intensity were used: below sensory threshold (below-threshold), sensory threshold (threshold), and above sensory threshold (above-threshold). The level of the intensity dial was gradually increased until each participant first perceived the sensory stimulus, which was set to the sensory threshold level (i.e., intensity level: 5). Subsequently, the above-threshold was set one step above the sensory threshold level (i.e., intensity level: 6) and the below-threshold was set one step below the sensory threshold level (i.e., intensity level: 4). According to a single blind study, the order of the intensities was chosen randomly to eliminate order effects. The participants were not provided with any information about the stimuli.

The experimental protocol is illustrated in Figure 2. In brief, during the pretest, the diameter of the blood vessels, BP, and PSV were measured for 30 s. In addition, the stimulation was initiated by simply turning on and immediately turning off the switch on the TENS equipment to eliminate placebo effects of TENS, and the pad was attached to the CCA location in the same manner as other stimulation measurements. TENS was then conducted with an intervention time of 15 s and an interstimulus interval of 5 s, as shown in Figure 2. This method, in which the off-time was shorter than the on-time, uses the effect of overlapping stimuli [35].

The PSV was measured throughout the 35 s of TENS intervention. The blood vessel diameters, BP, and PSV were measured 30 s following TENS. After a 5 min washout period, the experiment was repeated at different intensities. Throughout the experiment, participants were instructed not to speak while the PSV and CCA diameters were being measured.

### 2.3. Measurements

In this study, changes in PSV caused by the modulated stimulation intensity during short-term TENS intervention were assessed using Doppler ultrasound (RS85, Samsung Medison, Seoul, Republic of Korea). All data were obtained from the right CCA. Ultrasonographic measurements were taken 2 cm below the CCA bifurcation site, which was marked after identification using a diagnostic ultrasound. After the location was selected, the specimen was disinfected with an alcohol swab. To measure CCA diameter, axial images of the CCA were obtained using B-mode ultrasound. The PSV was acquired for 30 s in the sagittal plane of the CCA (Figure 3).

Using the Radiant DICOM Viewer (Mediant, Poznan, Poland), the outside diameters (OD) of the CCA during both vasoconstriction and vasodilation were measured from the acquired digital imaging and communications in medicine images, according to the arterial measurement standard [36].

To record the PSV in real time, the color Doppler mode was activated on an ultrasound monitor. A linear transducer array (LA2-14A; Samsung Medison, Seoul, Republic of Korea) with a frequency bandwidth of 2–14 MHz was applied to measure the PSV for 95 s. A total of 95 PSVs were obtained by recording one PSV per second. These PSV values were divided into the times before (30 s), during (35 s), and after (30 s) the TENS phases. The average PSV values were used to analyze the changes induced by TENS stimulation.

BP was measured using a wearable smartwatch (SM-R850, Samsung Electronics, Suwon, Republic of Korea), which allows measurement of BP within 30 s without compressing the blood vessels, as well as rapid remeasurement following TENS intervention. Accordingly, BP was obtained simultaneously with PSV and collected before and after TENS intervention. The measured data were automatically recorded using the firmware software (https://www.samsung.com/sec/support/model/SM-R850NZSAKOO) according to the manufacturer’s instructions.

#### Calculations of PWV and Stiffness Index (β)

Vascular stiffness was assessed using PWV, a principal clinical indicator that represents the elasticity of blood vessels, and is used to diagnose and predict arteriosclerosis and cardiovascular diseases [37]. PWV was calculated from blood stiffness and density using the following formula [2,38,39]:$$PWV=\sqrt{\beta \times \frac{DBP}{\left(2\times \rho \right)}}$$
where β is stiffness index (STIFF) and ρ is the blood density (1.05 g/cm^3^).

To calculate the local PWV, the CCA vasoconstriction and vasodilation diameters (VCD and VDD), SBP and DBP were measured. The diameters of the blood vessels were measured using ultrasound imaging, while BP was measured using a smartwatch. Subsequently, the STIFF (β) was computed using the following formula [40]:β=ln(SBP/DBP)×(VCD/(VDD−VCD)
where VCD and VDD are the diameters of the blood vessel during vasoconstriction and vasodilation, respectively.

### 2.4. Statistical Analysis

Statistical analyses were performed using Jamovi ver. 2.2.5 (The Jamovi Project (2021)). According to the central limit theorem over 30 participants, we assumed that the data would be normally distributed [41]. A one-factorial repeated-measures analysis of variance (ANOVA) was used to investigate the effects of the different TENS interventions (below-threshold, threshold, and above-threshold) among the three periods (before, during, and after TENS). The sphericity test was further performed using Mauchly’s test. If the sphericity assumption was not proven, the degrees of freedom were adjusted (ε > 0.75 = Huynh–Feldt; ε < 0.75 = Greenhouse–Geisser). The rate of change of PSV was expressed as the value derived by calculating the difference between the three variables (before, during, and after): (before−during)/before × 100 and (before−after)/before × 100. Post-hoc pairwise comparisons were further performed using Tukey’s honest significant difference (HSD) test, a representative method used to control Type 1 error rate in pair-wise comparisons. A paired-sample *t*-test was further applied to analyze significant differences between the PWV (before and after) values. Statistical comparison of the difference in PWV (before and after) of TENS intervention (below-threshold, threshold, and above-threshold) was performed using one-factorial repeated-measure ANOVA. The standard criterion of statistical significance (*p* < 0.05) was applied for all analyses.

## 3. Results

### 3.1. Changes in PSV in Response to TENS Intervention at Each Stimulation Intensity

The changes in PSV following TENS intervention at each of the three intensities are shown in Table 2. After the TENS intervention at all intensities, a decreasing trend in the average PSV was observed. First, for the above-threshold level, the PSV before stimulation was 96.5 cm/s. However, during and after the stimulation, the PSV decreased to 94.4 cm/s and 94.6 cm/s respectively, both lower than the values before the intervention. At the threshold level, the average PSV was 93.7 cm/s prior to stimulation, remaining at 93.7 cm/s during the intervention, but decreasing to 92.8 cm/s after stimulation. However, these changes in PSV for both the threshold and above-threshold levels did not appear to induce any significant difference in the response to the intervention (*p* = 0.100 and *p* = 0.708). In contrast, a decrease in the PSV to the below-threshold level was observed following the intervention (*p* = 0.028). The PSV before stimulation at the below-threshold level was 94.2 cm/s, which decreased to 92.8 cm/s during the intervention and continued to decrease to 91.4 cm/s after stimulation. In particular, we observed a significant decrease after the intervention compared with before the intervention (*p* = 0.031).

### 3.2. Rate of Change in PSV Following Intervention at Each Stimulation Intensity

With the exclusion of the below-threshold condition, significant changes were not observed before, during, or after the TENS phases. The degree of change in the PSV in response to stimulation was quantified, and the amount of change among different intensities was compared. The difference in PSV during stimulation compared with that before stimulation is shown in Table 3. Among the stimuli, at the above-threshold level, the PSV showed the largest rate of change, averaging at 2.38%. At the threshold level, the average PSV rate of change increased by 0.65% during stimulation. However, at the below-threshold level, the average PSV rate of change decreased by 0.82% during the stimulation. Despite these changes, no significant difference was noted in the rate of change in the PSV among the different stimuli (*p* = 0.219).

The differences in the PSV before and after stimulation also did not reach statistical significance (see Table 3). After stimulation, the PSV tended to decrease at all intensities. A 1.30% decrease in the PSV rate of change was observed the above-threshold level. At the threshold level, the PSV rate of change decreased by 0.12%. At the below-threshold level, the PSV rate of change exhibited an average decrease of 2.23%. Following the intervention, the rate of change in PSV decreased the most, on average, to the below-threshold level; however, it did not reach statistical significance (*p* = 0.536).

### 3.3. Changes in PWV in Response to TENS Intervention at Each Stimulation

No significant changes were observed in PWV at any intensity (see Table 4). However, the PWV above-threshold level increased by 0.34 m/s from 4.42 m/s to 4.77 m/s, the largest increase among all intensities. In contrast, PWV decreased after intervention at both the threshold and below-threshold levels. At the threshold level, PWV decreased by 0.14 m/s from 4.48 m/s to 4.34 m/s. At the below-threshold level, PWV decreased by 0.02 m/s from 4.32 m/s to 4.30 m/s.

## 4. Discussion

Overall, this study demonstrated that electrical stimulation induced a significant decrease in the flow velocity in the CCA. The PSV tended to decrease at all intensities following TENS stimulation (see Table 2). In particular, a significant decrease in PSV at the below-threshold level was observed following TENS intervention. Conversely, significant changes in PWV for assessing vascular flexibility were not observed at any electrical stimulation intensity, although an increase in PWV was observed at the above-threshold level, indicating that high-intensity electrical stimulation is likely to cause vascular stiffness, even if it does not cause significant changes in the CBF.

The cardiovascular effects induced by TENS have been reported since the 1970s, with the underlying physiological mechanisms being attributed to the release of vasodilators and the modulation of the sympathetic nervous system [42,43]. According to previous studies, when TENS is applied to the paravertebral ganglion region, low-frequency stimulation decreases the activity of the sympathetic nervous system, while high-frequency stimulation increases the activity of the sympathetic nervous system [24]. Therefore, TENS also acts on the sympathetic nervous system branched from the CCA. According to the ANS control of the cardiovascular system, most blood vessels are controlled by the sympathetic nerves, while vasoconstriction is regulated through the action of norepinephrine [44,45]. Therefore, TENS affecting the CCA can stimulate the sympathetic nerves of blood vessels.

The observation of reduced PSV to the below-threshold level could indicate that low-intensity electrical stimulation may trigger Nitric Oxide (NO) release, a known mediator of vasodilation [46]. As shown in previous studies, electrical stimulation affects arterial elasticity by inducing the release of the vasodilator NO from vascular endothelial cells [46,47]. The ion flow in the cell membrane caused by such electrical stimulation can also trigger NO release. In particular, calcium ion channels activated by electrical stimulation can decrease potassium permeability and significantly alter the membrane potential, resulting in the release of NO [48]. In addition, various clinical studies have inferred the effect of NO on blood flow changes in response to electrical stimulation [43,49]. The released NO triggers vasodilation by relaxing the smooth muscles of the blood vessels. In fact, following low-intensity electrical stimulation, the tone of the vascular smooth muscle decreased owing to NO release, while both the heart–ankle and brachial–ankle PWVs were significantly decreased due to vasodilation [43]. Consistent with the results of prior studies, the present study further showed a tendency for PWV to decrease on average, while the vascular flexibility increased following TENS intervention. Short-term low-intensity electrical stimulation was confirmed to be effective at reducing blood flow velocity by affecting blood vessel flexibility.

At both the threshold and above-threshold levels, TENS interventions tended to decrease PSV following the intervention, although the difference did not reach significance. This phenomenon appears to be caused by threshold level electrical stimulation that excessively stimulates the sympathetic nerves innervating the blood vessels. According to previous studies, TENS can affect reflexes related to the ANS, which represents a key mechanism of the central nervous system in regulating the cardiovascular system [50]. Vasoconstriction is further controlled by sympathetic tone, while vasodilation is induced by the withdrawal of sympathetic tone [44]. Therefore, excessive electrical stimulation can cause vasoconstriction in the CCA, thereby restricting blood flow. In the present study, the greatest decrease in the PSV was observed during stimulation at the above-threshold level. However, the rate of decrease in the PSV decreased following stimulation, probably due to the activation of sympathetic nerves in blood vessels by electrical stimuli, which induces vasoconstriction and inhibits significant changes in blood flow velocity. Additionally, TENS interventions at both the threshold and below-threshold levels were observed to have some effect on vascular stiffness; however, this impact was not significant. Among the three stimulation intensities, PWV tended to increase only the above-threshold level. This increase in PWV above-threshold level appears to be due to the effect of high-intensity electrical stimulation, which increases the activity of sympathetic nerves innervating the blood vessels, leading to vasoconstriction. In fact, according to Ferrell & Khoshbaten (1989) [51], when electrical stimulation is applied to the blood vessels, it can induce vasoconstriction due to the action of noradrenaline on α1-adrenoceptors. Norepinephrine, released from the postganglionic sympathetic nerves of the blood vessels, binds to α1 or α2 adrenergic receptors located in the vascular smooth muscle cells. This increases the release of Ca^2+^ within the cells, thereby inducing the contraction of the vascular smooth muscle [52,53]. Additionally, studies have reported that high intensity electrical stimulation can cause vasoconstriction by directly stimulating vascular smooth muscle [54]. Therefore, electrical stimulation at the above-threshold level likely induces more sympathetic nerve activity than other intensities, thereby affecting the tendency for PWV to increase due to vasoconstriction.

## 5. Limitations

This study has some limitations. First, implying clinical effects based on the study design was challenging. Indeed, this was only a preliminary study aiming to confirm the effect of CBF regulation through TENS. Therefore, it was conducted in healthy young adults in their 20s; however, further research is needed to confirm the clinical implications of TENS. The results from this study confirmed the possibility of TENS. If physiological changes in blood vessels were observed in young adults with good vascular elasticity, it could be expected that greater effects would be observed in patients with cardiovascular disease who have high vascular stiffness and high blood pressure. For example, in patients with cardiovascular diseases, such as hypertension and arteriosclerosis, where the average blood flow velocity is elevated [55,56], a more distinct effect of decreased blood flow velocity is anticipated when TENS intervention is performed. Furthermore, in this study, the electrodes were attached to the CCA location at the front of the neck, which is generally avoided, in contrast to the conventional application method of TENS. To minimize the side effects caused by electrical stimulation, the parameters were set to the minimum frequency and pulse width of the TENS device rather than at the stimulation variables generally used in TENS interventions. Therefore, further research is required to identify the optimal electrical stimulation parameters for blood vessels. Finally, various potential confounding variables related to the cardiovascular system should be considered for future studies targeting cardiovascular patients.

## 6. Conclusions

We examined the effects of TENS on cerebral blood vessels at threshold, below-threshold, and above-threshold levels. Overall, our results showed that the blood flow velocity significantly decreased below-threshold level after the intervention. In contrast, although a tendency toward decreased blood flow velocity was observed at and above-threshold levels, no significant differences were observed. This appears to be due to the effect of a vasodilation mediator, which is activated only when electrical stimulation is at the below-threshold level. At above-threshold levels, vasoconstriction due to sympathetic nerve stimulation of the blood vessels appears to interfere with the decrease in blood flow velocity more than the release of vasodilation mediators. Consequently, we discovered that a brief 35 s electrical stimulation at the below-threshold level does not affect vascular biophysical function; however, it was found to have an immediate effect in reducing blood flow velocity. The electrical stimulation technique used in this study lowers the vascular resistance and contributes to the prevention of blood vessel damage due to rapid blood flow velocity, such as hypertension and arteriosclerosis. In addition, this reduction in the risk of vascular damage can help to maintain smooth CBF and ultimately prevent the development of brain diseases.

## Figures and Tables

**Figure 1 healthcare-12-01908-f001:**
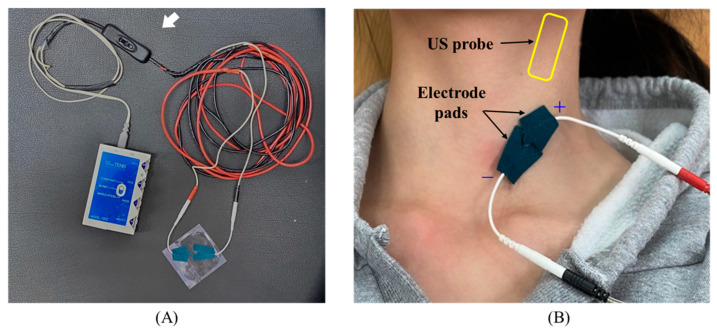
Images of the modified TENS system and electrode pads. (**A**) TENS connected with on/off button (arrow). The on/off button was applied to ensure more accurate control of the stimulus intervention time. (**B**) An image of the attached location of electrode pads and the position of the ultrasound (US) probe on neck.

**Figure 2 healthcare-12-01908-f002:**
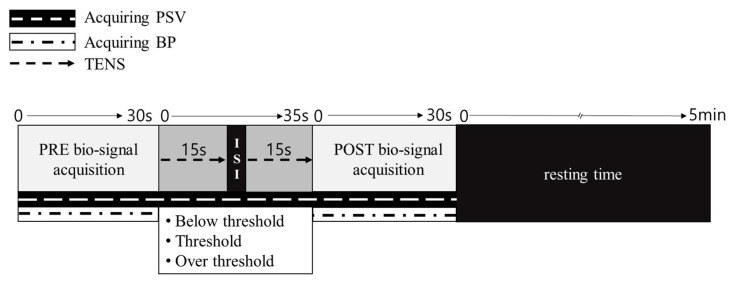
Biosignal acquisition protocols in the PRE and POST TENS. Biosignals represent the measurements of peak systolic velocity (PSV) and blood pressure (BP). ISI, interstimulus interval.

**Figure 3 healthcare-12-01908-f003:**
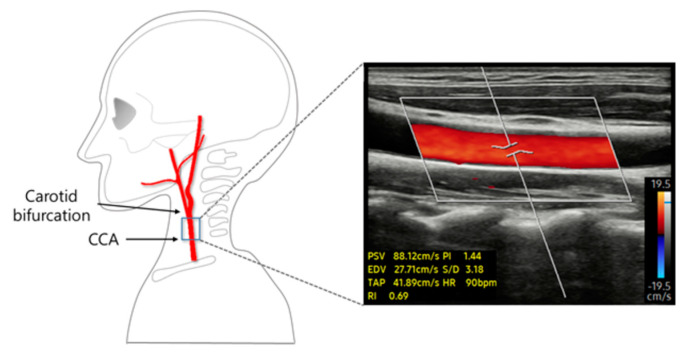
A representative image of the blood flow velocity image of the CCA, measured using color doppler mode ultrasound.

**Table 1 healthcare-12-01908-t001:** Participants’ general characteristics and vital signs at rest.

Variables	Total (n = 31)	Male (n = 14)	Female (n = 17)
Age (years)	22.71 ± 1.79	23.21 ± 2.19	22.29 ± 1.31
Height (cm)	167.94 ± 6.44	173.00 ± 4.31	163.77 ± 4.66
Weight (kg)	61.94 ± 9.07	68.71 ± 8.30	56.35 ± 5.00
SBP (mmHg)	114.55 ± 7.06	115.43 ± 6.99	113.82 ± 7.25
DBP (mmHg)	68.74 ± 5.43	67.43 ± 4.86	69.82 ± 5.78
HR (bpm)	71.48 ± 8.04	71.64 ± 8.81	71.35 ± 7.62

Abbreviations: DBP, diastolic blood pressure; HR, heart rate; SBP, systolic blood pressure.

**Table 2 healthcare-12-01908-t002:** Comparison of the PSV (cm/s) changes before, during, and after TENS at each intensity level.

Repeated Measures ANOVA					Post-Hoc Comparisons (Tukey)		
Variables		Mean ± SD	F	*p*	Variables	t	*p*
Above-threshold	Before	96.5 ± 13.9	2.61	0.100	Before − During	3.47	0.005
	During	94.4 ± 13.8			Before − After	1.50	0.305
	After	94.6 ± 11.9			During − After	−0.386	0.921
Threshold	Before	93.7 ± 13.9	0.26	0.708	Before − During	−0.03	0.999
	During	93.7 ± 14.2			Before − After	0.48	0.880
	After	92.8 ± 14.2			During − After	0.82	0.694
Below-threshold	Before	94.2 ± 13.6	4.46	0.028 *	Before − During	2.35	0.064
	During	92.8 ± 12.3			Before − After	2.68	0.031 *
	After	91.4 ± 11.7			During − After	1.32	0.396

Abbreviations: ANOVA, analysis of variance; SD, standard deviation. * *p* < 0.05.

**Table 3 healthcare-12-01908-t003:** Comparison of the before−after and before−during change rates of PSV at each stimulation intensity.

Repeated Measures ANOVA				
Outcome Variables	Variables	ΔDIFF ± SD (%)	F	*p*
Before − During	Above-threshold	2.38 ± 3.76	1.56	0.219
	Threshold	−0.65 ± 8.91		
	Below-threshold	0.82 ± 3.45		
Before − After	Above-threshold	1.30 ± 7.38	0.63	0.536
	Threshold	0.12 ± 12.80		
	Below-threshold	2.23 ± 6.34		

Abbreviations: ANOVA, analysis of variance; DIFF, difference value; SD, standard deviation.

**Table 4 healthcare-12-01908-t004:** Comparison of PWV (m/s) between before and after TENS.

			Paired *t*-Test			Repeated Measures ANOVA		
	Variables		Mean ± SD	t	*p*	ΔDIFF (Mean ± SD)	F	*p*
PWV	Above-threshold	Before	4.42 ± 1.12	−1.29	0.207	−0.34 ± 1.44	0.956	0.391
		After	4.77 ± 1.73					
	Threshold	Before	4.48 ± 1.07	0.67	0.510	0.14 ± 1.12		
		After	4.43 ± 0.68					
	Below-threshold	Before	4.32 ± 1.03	0.09	0.929	0.02 ± 1.22		
		After	4.30 ± 0.74					

Abbreviations: ANOVA, analysis of variance; DIFF, difference value; PWV, pulse wave, velocity; SD, standard deviation.

## Data Availability

The ethical approval conditions prohibit the public storage of anonymized research data. However, individuals seeking access to the data may do so after completing a formal data sharing agreement and following ethical procedures.

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
