# Peer review of "Regulation of Cerebral Blood Flow Velocity by Transcutaneous Electrical Nerve Stimulation: A Preliminary Study"

_healthcare, 2024, doi:10.3390/healthcare12191908_

Round 1
Reviewer 1 Report
Comments and Suggestions for Authors
Althoug the sample size is small, the study is novel. It merits reading. However, English writing should be improved.
Comments on the Quality of English LanguageThe English writing should be checked by a native English-speaker.
Reviewer 2 Report
Comments and Suggestions for Authors
This study investigated the effects of transcutaneous electrical nerve stimulation (TENS) on blood flow in the CCA. The results indicate that below-threshold stimulation significantly reduces blood flow velocity without affecting arterial stiffness, suggesting its potential as a short-term method to lower vascular resistance and prevent damage from rapid blood flow. In contrast, above-threshold stimulation may increase vascular tone and trigger vasoconstriction, preventing a decrease in blood flow velocity.
Reviewer Comment (Major): The manuscript frequently uses the term 'blood flow velocity,' but it is important to note that 'blood flow' and 'velocity' are distinct units and should not be used interchangeably. Blood flow typically refers to the volume of blood passing through a vessel per unit of time (e.g., mL/min), whereas blood flow velocity refers to the speed at which blood moves through the vessel (e.g., cm/s). I recommend that the authors clarify the terminology throughout the manuscript and ensure that the correct is used in each context. For example, if the authors are discussing the rate of blood movement, 'velocity' should be used; if referring to the amount of blood passing through a vessel, 'flow' should be used instead (please read the article PMID: 35839156)
Reviewer Comment (Major): CCA diameter alone is not sufficient to directly calculate PWV, it can be used as part of a more comprehensive calculation involving changes in diameter, pressure, and other related factors. Please at least add this to limitation
Reviewer Comment (Major): While I appreciate the effort invested in this study, the relatively small sample size raises concerns about whether the study is adequately powered to detect meaningful effects. Therefore, I request a more detailed justification for the chosen sample size, including a power analysis or relevant calculations. Please provide a comprehensive response addressing the effect size calculation and the minimum sample size required to reliably detect the effect
Comments on the Quality of English LanguageEnglish quality is good
Reviewer 3 Report
Comments and Suggestions for Authors
This study investigates the effects of different intensities of Transcutaneous Electrical Nerve Stimulation (TENS) on cerebral blood flow (CBF) and vascular stiffness, focusing on the common carotid artery (CCA). Thirty-one healthy adults participated in the study, with TENS applied at below-threshold, threshold, and above-threshold intensities. The study found that below-threshold TENS significantly decreased the peak systolic velocity (PSV) in the CCA, while no significant changes were observed in pulse wave velocity (PWV) for any intensity level. I have several comments, both major and minor, for authors to consider:
Major concerns:
1. The study uses a small sample size (n=31) of healthy adults in their twenties, which limits the generalizability of the findings. Perhaps the authors could specify in the title of the manuscript, abstract, and methodology that it is a preliminary or pilot study that is sought to inform future study design.
2. The lack of diverse age groups and individuals with varying health statuses is a significant limitation, especially given the clinical implications related to cardiovascular health.
3. Although the study mentions randomizing the order of stimulation intensities, there is no mention of blinding. Lack of blinding could introduce bias, as participants' awareness of the stimulation intensity might influence their physiological responses.
4. The absence of a control group (e.g., sham stimulation) makes it difficult to attribute observed changes solely to TENS intervention rather than placebo effects or natural variability in physiological measures.
5. Page 5: The TENS intervention duration of 35 seconds may be too short to observe meaningful physiological changes, particularly in vascular stiffness, which might require more prolonged exposure to stimulation.
6. Page 8 (Mechanistic Explanation): The proposed mechanisms, such as the release of nitric oxide (NO) and its effect on vascular tone, are not directly measured or validated in this study. Without biochemical or molecular data, these claims remain speculative.
7. Page 11: The discussion lacks depth in connecting findings to existing literature about TENS effects on vascular parameters. More thorough integration with prior studies could strengthen the scientific rationale and interpretation of results.
8. Given the small sample size, the study may lack the statistical power to detect significant differences, particularly for PWV measurements where no significant changes were observed. It is not clear if power analysis was performed - as this would clarify whether the study is adequately powered.
9. The use of multiple ANOVAs and post-hoc tests increases the risk of Type I errors. The study does not mention corrections for multiple comparisons, which could lead to inflated significance levels.
10. While the study suggests potential neuroprotective effects of low-intensity TENS, the clinical significance of the observed changes in PSV is unclear. The manuscript does not discuss whether these changes are large enough to impact clinical outcomes, such as stroke prevention.
11. There are also some safety concerns that merit consideration. The authors briefly mention discomfort with TENS application (Page 3) to the neck area but do not thoroughly address safety concerns, especially given the potential for adverse effects related to carotid sinus stimulation.
12. The study does not account for potential confounding variables such as participants' baseline cardiovascular fitness or stress levels, which could influence CBF and vascular responses.
Minor concerns:
13. With regards to data availability (Page 10), the authors note that data are available upon request but do not specify any conditions or procedures for access, which could hinder reproducibility and verification of findings.
14. There are several instances of grammatical errors and typos throughout the manuscript, such as "stimulus" instead of "stimuli" (Page 5, Line 238), and missing articles or prepositions. These issues can distract readers and should be addressed to improve readability.
15. There is some inconsistency in the terminology used. For instance, the terms "below-threshold," "threshold," and "above-threshold" are used variably (e.g., "below threshold"), which might confuse readers. Standardizing these terms throughout the manuscript would enhance clarity.
16. The statistical analysis section would benefit from more detail about the assumptions checked for ANOVA and any post-hoc tests used. It should explicitly mention any corrections made for multiple comparisons.
17. The title is somewhat lengthy and may benefit from simplification or rephrasing for clarity. It currently provides a comprehensive overview of the study's focus but could be more concise.
Comments on the Quality of English LanguageSeveral grammatical errors and typos occur throughout the manuscript, which should be addressed to improve readability.
Round 2
Reviewer 2 Report
Comments and Suggestions for Authors
Thank you for your response.
Reviewer 3 Report
Comments and Suggestions for Authors
No further comments
Comments on the Quality of English LanguageOk